# Plasma microRNA Profiling in Type 2 Diabetes Mellitus: A Pilot Study

**DOI:** 10.3390/ijms242417406

**Published:** 2023-12-12

**Authors:** Ziravard N. Tonyan, Yury A. Barbitoff, Yulia A. Nasykhova, Maria M. Danilova, Polina Y. Kozyulina, Anastasiia A. Mikhailova, Olga L. Bulgakova, Margarita E. Vlasova, Nikita V. Golovkin, Andrey S. Glotov

**Affiliations:** 1Department of Genomic Medicine, D.O. Ott Research Institute of Obstetrics, Gynecology and Reproductology, 199034 St. Petersburg, Russia; ziravard@yandex.ru (Z.N.T.); barbitoff@bk.ru (Y.A.B.); yulnasa@gmail.com (Y.A.N.); elenamariamassa@gmail.com (M.M.D.); polykoz@gmail.com (P.Y.K.); anamikhajlova@gmail.com (A.A.M.); o.bulgakova1310@gmail.com (O.L.B.); 2St. Martyr George City Hospital, 194354 St. Petersburg, Russia; vlasovaritafi@gmail.com (M.E.V.); golovkin1983@yandex.ru (N.V.G.)

**Keywords:** type 2 diabetes mellitus, obesity, miRNA, miR-496, miR-5588-5p, miR-125b-2-3p, miR-1284, miR-144-3p, miR-99a-5p, obesity

## Abstract

Type 2 diabetes mellitus (T2D) is a chronic metabolic disease characterized by insulin resistance and β-cell dysfunction and leading to many micro- and macrovascular complications. In this study we analyzed the circulating miRNA expression profiles in plasma samples from 44 patients with T2D and 22 healthy individuals using next generation sequencing and detected 229 differentially expressed miRNAs. An increased level of miR-5588-5p, miR-125b-2-3p, miR-1284, and a reduced level of miR-496 in T2D patients was verified. We also compared the expression landscapes in the same group of patients depending on body mass index and identified differential expression of miR-144-3p and miR-99a-5p in obese individuals. Identification and functional analysis of putative target genes was performed for miR-5588-5p, miR-125b-2-3p, miR-1284, and miR-496, showing chromatin modifying enzymes and apoptotic genes being among the significantly enriched pathways.

## 1. Introduction

Type 2 diabetes mellitus (T2D) is a chronic metabolic disease characterized by insulin resistance and β-cell dysfunction resulting in persistent hyperglycemia. Patients with T2D are at increased risk of microvascular and macrovascular complications, such as retinopathy, peripheral neuropathy, nephropathy, atherosclerotic cardiovascular, and cerebrovascular disease. Also, arterial hypertension, obesity, and dyslipoproteinemia often accompany T2D [1]. The wide range of complications and the predicted increase in the prevalence of T2D make presymptomatic testing and early diagnosis of diabetes extremely important. Incomplete understanding of molecular mechanisms underlying T2D, as well as a lack of optimal early biomarkers, is one of the few challenges that researchers face in developing approaches to presymptomatic testing. A perfect biomarker, according to the SMART concept proposed by Shehabi and colleagues, should meet the following criteria: it is Sensitive and Specific, Measurable, Available and Affordable, Responsive and Reproducible, allowing for Timely repeated measurements [2,3]. MicroRNAs (miRNAs) are a potential biomarker for the identification of high-risk groups of developing T2D.

MiRNAs are small (~22 nt) non-coding RNAs involved in transcriptional and post-transcriptional regulation of gene expression [4] by binding to a specific site in the 3’UTR (untranslated region) of the target mRNA and thereby inducing mRNA deadenylation and decapping [5]. MiRNAs could potentially become early or presymptomatic biomarkers due to their role in the initial stages of T2D pathogenesis. Even low-abundant miRNAs can be detected easily with the amplification stage, unlike proteins and metabolites [6]. In addition, the miRNA stability enables expression measurement, which is easier when compared to the target mRNA [7]. Techniques that are currently in use for miRNA expression analysis include quantitative reverse transcription PCR (RT-qPCR), expression arrays, and next generation sequencing (NGS). While expression arrays and NGS are mainly used in scientific research, RT-qPCR is actively used both in clinical practice and in scientific experiments as it is fast, affordable, modern, and accurate.

To date, altered expression of numerous miRNAs has been demonstrated in patients with T2D in several studies. Since miRNA meets most of the requirements for an ideal biomarker, the aim of the present study was to analyze the spectrum of differentially expressed miRNAs in the blood plasma of T2D patients, to find differentially expressed miRNAs associated with T2D and suitable for further validation on an extended sample, and to analyze the potential target genes of these miRNAs and the processes in which they are involved. 

## 2. Results

### 2.1. Identification of Differentially Expressed Circulating miRNA

MiRNA expression profiles were determined in blood plasma samples of T2D patients and healthy controls using the NGS approach. More than 1 million miRNA reads were mapped for each sample. Analysis of total miRNA expression profiles using principal component analysis (PCA) showed lack of clear separation and clustering of control and T2D samples (Figure 1a), which suggested the existence of factors other than disease status affecting the miRNA expression. Nevertheless, the differential expression analysis revealed 229 miRNA with significant differences in expression levels between control samples and patients with T2D (the entire list is provided in Appendix A). Of these, 192 miRNA showed more than two-fold expression change between study groups, with 147 of them upregulated and 45 downregulated in T2D patients. Notably, none of the most studied miRNA markers of T2D (hsa-miR-126, hsa-miR-375, hsa-miR-34a, hsa-miR-29a, hsa-miR-144) were found among the differentially expressed ones [8,9,10,11,12].

We next performed an additional differential expression analysis grouping patients by T2D risk factors, such as obesity and family history of diabetes, since obesity and T2D in close relatives were known risk factors for T2D [13,14]. To analyze these factors, the following groups of participants were formed: (a) 25 participants with a family history of T2D and 41 individuals without close relatives with T2D; (b) 23 obese patients with BMI ≥ 30 and 43 subjects with BMI < 30. Differentially expressed miRNA were detected in all such analyses (Figure 1b,c); however, far fewer miRNAs were identified as differentially expressed for obesity (38) and only a single such miRNA was found for family history of T2D. All but two of the obesity-associated miRNAs presented expectedly also among the ones differentially expressed in T2D (Figure 1c). The only obesity-specific miRNAs were miR-144-3p and miR-99a-5p. Taken together, these results suggest that the development of T2D has a much larger impact on the miRNA expression profile compared to obesity, and that miRNAs other than the most widely studied ones contribute to the observed differences. Hence, we went on to select several of the identified differentially expressed miRNAs for further experimental validation and use as potential T2D markers.

### 2.2. RT-qPCR Verification

MiR-496, miR-5588-5p, miR-125b-2-3p, and miR-1284 were selected for RT-qPCR quantification to verify the reliability of the results obtained by high-throughput sequencing as the most promising based on the results of filtering by adjusted *p*-value, |log2FC|, and the proportion of samples with zero expression.

To validate that the set of selected miRNA (miR-5588-5p, miR-125b-2-3p, miR-1284, and miR-496) effectively separates T2D patients from the control samples, we constructed a machine learning model for the prediction of the patient’s phenotype based on the expression of four identified miRNA. Such a model showed high predictive power (ROC/AUC = 0.94 ± 0.10 in three-fold cross-validation).

MiR-361-5p demonstrated stable expression levels in both T2D patients and controls and was chosen as an endogenous control. MiR-5588-5p, miR-125b-2-3p, miR-1284 were upregulated, and miR-496 was downregulated in T2D patients (log 2 fold change was 4.99, 4.85, 4.64, and −2.0, respectively) according to the results of NGS. The verification results were consistent with those of NGS and showed reduced expression of miR-496 (*p* = 0.036) and a minor trend towards increased expression of miR-5588-5p (*p* = 0.06), miR-125b-2-3p (*p* = 0.08), and miR-1284 (*p* = 0.14) in T2D patients (Figure 2).

### 2.3. Identification and Functional Analysis of Putative miRNAs Target Genes

After the observed differential expression of selected miRNAs was experimentally validated, we next turned to investigation of their potential functional effects. To do so, we retrieved known target genes of these miRNA using the miRTarBase v8. In total, 288 target genes were determined. We next analyzed this set of target genes using Gene Ontology (GO) term enrichment analysis (see Section 4). This analysis revealed only two biological process terms reaching statistical significance (adjusted *p*-value < 0.1), including response to ketone and response to dexamethasone (Figure 3). Enrichment analysis of the same gene set using the Molecular Signatures Database (MSigDB) canonical pathway set discovered more significant hits, with chromatin modifying enzymes and apoptotic pathways genes among the significantly enriched pathways.

Of particular interest were 16 target genes differentially expressed in T2D patients according to previously published transcriptomic studies and also located at the loci associated with T2D according to GWAS data [15] (*ACVR1C*, *ATXN7*, *DCUN1D4*, *GIN1*, *GOLGA7*, *GTF3C2*, *HMG20A*, *HMGB1*, *INTS8*, *KIF11*, *RNF6*, *SBN1*, *SDC2*, *SSR1*, *UBE3C*, *ZFP36L2*).

## 3. Discussion

Altered miRNA expression was demonstrated in various diseases, including cancer, viral diseases, immune-related, endocrine, and neurodegenerative diseases [6]. Over the past decade, many studies have been conducted on the analysis of miRNA expression in T2D using qPCR, expression arrays, and NGS approaches to shed light on the molecular basis of this disease and to find the effective biomarkers for the risk assessment of T2D and its complications [16]. Differentially expressed miRNAs were found in striated muscle, pancreatic islet cells, adipose tissue, whole blood, serum, and plasma [16,17,18,19,20,21,22,23,24]. While striated muscle, pancreatic islet cells, and adipose tissue are undoubtedly of scientific interest for the study of T2D, they are difficult to access and are not suitable for use as a biomaterial for assessing the risk of T2D in routine clinical practice. Plasma, serum, and whole blood are more easily accessible for testing and can be used for the search of T2D biomarkers.

In the present pilot study, we compared miRNA expression profiles in plasma samples from T2D patients and healthy controls using the NGS method to find differentially expressed miRNAs associated with T2D suitable for further validation on extended sample. A total of 229 differentially expressed miRNAs were identified. For four of them with the most altered expression levels (miR-5588-5p, miR-125b-2-3p, miR-1284, and miR-496), a simultaneous change in expression was confirmed by RT-qPCR. To our knowledge, this is the first time that dysregulation of miR-496, miR-125b-2-3p, miR-1284, and miR-5588-5p has been demonstrated in the plasma of patients with T2D. To date, differential expression of these miRNAs was demonstrated in conditions such as mild cognitive impairment due to Alzheimer’s disease (miR-5588-5p), hepatocellular carcinoma (miR-125b-2-3p), gastric cancer (miR-1284), and osteosarcoma (miR-1284, miR-496) [25,26,27,28,29]; however, no simultaneous changes in the level of these miRNAs are known. Therefore, the detected miRNAs are of particular interest since they are rarely differentially expressed in other conditions, making them quite specific, especially when analyzed simultaneously. Expression changes were verified using RT-qPCR, but further validation is required in a larger sample. The participation of the detected miRNAs in the development of T2D has not been sufficiently studied to date; however, some mechanisms can be assumed based on the nature of their involvement in the pathogenesis of other conditions. 

MiR-496 was previously studied mainly in patients with malignant tumors. Previous research demonstrated a significant role of miR-496 in the processes of cell proliferation and differentiation. For example, a suppression of tumor cell proliferation through AKT/mTOR and PI3K/AKT signaling pathways is well known [30,31]. It can be assumed that this miRNA could be also involved in the regulation of β-cell proliferation. Our results are consistent with the available data on reduced expression of miR-496 in patients with T2D [32]. Downregulation of miR-496 was previously demonstrated in peripheral blood mononuclear cells (PBMCs), while miR-496 levels were reduced in plasma in the present study. One of the possible mechanisms for the participation of miR-496 in the pathogenesis of T2D is a persistent activation of mTOR complex caused by downregulated miR-496 expression, since an inverse correlation between miR-496 expression and the amount of mTOR proteins in PBMC was shown [33]. mTOR is a conserved serine/threonine kinase complex involved in the metabolism of proteins, lipids, carbohydrates, nucleotides, and regulating cell survival and cytoskeleton remodeling. Short-term activation of mTOR is shown to stimulate β-cell proliferation and increase β-cell mass, whereas permanent activation has a negative impact on β-cell mass and function [34]. 

MiR-125b-2-3p expression was mainly studied in patients with hepatocellular carcinoma, ischemic stroke, and also was considered as a predictor of the therapy effectiveness in colorectal cancer [29,35,36]. This miRNA can act both as an oncogene and as a tumor suppressor in tumors of different origin. It can be also involved in proliferation, differentiation, migration, and invasion of tumor cells [37]. The participation of miR-125b-2-3p in T2D pathogenesis can be explained through AMP-activated protein kinase (AMPK) way. AMPK is known to be the key regulator of glucose metabolism in insulin-dependent tissues (skeletal muscles, liver, and adipose tissue) and β-cells; therefore, AMPK dysregulation in β-cells causes impairment of glucose homeostasis [38]. Cheung and colleagues previously showed that β-cell-specific inactivation of AMPK in mice was associated with miR-125b overexpression, concluding that AMPK may act as a negative regulator of miR-125b expression [39]. More recently, in vitro and in vivo studies demonstrated that upregulation of miR-125b caused by AMPK repression in islet cells impairs glucose-stimulated insulin secretion and thus leads to hyperglycemia and glucose intolerance. The authors hypothesized these changes are caused by targeting *M6PR* and *MTFP1* genes associated with mitochondrial and lysosomal functions [40]. MiR-125b overexpression was demonstrated not only in islet cells, but also in other tissues. MiR-125b level was upregulated in visceral adipose tissue and PBMCs of T2D and prediabetic patients compared to healthy ones [41,42]. The differential expression of miR-125b in adipose tissue may be associated with its suppression of adipogenesis of brite adipose tissue by modulating the expression of mitochondrial protein UCP1 [43]. BMI also plays a crucial role in miR-125b expression-regulated glucose metabolism: miR-125b knockout reduced insulin sensitivity and, consequently, glucose utilization in mice with high-fat diet-induced obesity [44].

The dysregulation of miR-1284 expression has not yet been described in T2D patients, but it was studied in patients with breast, gastric, and lung cancer [45,46,47]. To date, one of the main and essential functions of miR-1284 is known to be participation in the regulation of cell viability and apoptosis. Overexpression of miR-1284 was shown to significantly inhibit the proliferation, migration, and invasion of breast cancer cells. One of the possible mechanisms is targeting *ZIC2*, a potential oncogene for many types of cancer [47,48]. MiR-1284 also reduced cell viability and stimulated apoptosis in ovarian cancer cells by affecting p27 and via the PI3K/Akt pathway, both of which play a significant role in cell cycle regulation [49]. When evaluating the viability and proliferation of lung cancer cells, a similar tendency was demonstrated. Overexpression of miR-1284 led to the stimulation of apoptosis, a decrease in cell viability and proliferation by activating p27 [46]. One of the possible mechanisms for the miR-1284 involvement in T2D development is also activation of p27, since it regulates the proliferation of β-cells, and an increase in p27 expression reduces β-cell mass, resulting in impaired glucose tolerance [50,51].

MiR-5588-5p has not been sufficiently studied to date; its expression was examined only in the cerebrospinal fluid of patients with a mild cognitive disorder due to Alzheimer’s disease and served as a marker of cognitive deterioration [28]. To date, there is no information about experimentally confirmed target genes for this miRNA. Further studies are required to evaluate the expression levels of miR-5588-5p in T2D patients. Possible mechanisms of the detected miRNAs participation in T2D pathogenesis, as well as tissues and biological fluids in which they are expressed, are summarized in Table 1. 

In the present study, miRNA expression profiles were also compared in plasma samples from obese and non-obese patients. The results demonstrated differential expression of miR-144-3p and miR-99a-5p in obese patients. Several mechanisms are known for miR-144-3p to promote adipogenesis: from suppressing the *FOXO1* and reducing its regulation of adiponectin to stimulating differentiation of adipocytes by direct targeting Klf3 and CtBP2, protective factors against obesity [52,53]. MiR-99a-5p was also reported to be negatively correlated with obesity [54,55].

According to functional analysis, the most enriched molecular pathways of T2D pathogenesis include chromatin modifying enzymes and apoptotic pathway genes. The target genes identified during the functional analysis partly overlapped with those differentially expressed in T2D patients based on the results of RNA-seq. According to GWAS results, 16 of them (*ACVR1C*, *ATXN7*, *DCUN1D4*, *GIN1*, *GOLGA7*, *GTF3C2*, *HMG20A*, *HMGB1*, *INTS8*, *KIF11*, *RNF6*, *SBN1*, *SDC2*, *SSR1*, *UBE3C*, *ZFP36L2*) were also located in loci associated with T2D. The association of these genes with cellular processes such as cell-cycle control and apoptosis is consistent with the results of studies demonstrating the significance of these processes in the pathogenesis of T2D and its complications. Activation of cell cycle regulatory genes predicts T2D and correlates with β-cell proliferation, as shown earlier [56]. It is also known that cell-cycle dysregulation is an important link in the pathogenesis of diabetic kidney disease [57]. Β-cell apoptosis also plays a significant role in the pathogenesis of T2D along with obesity-associated insulin resistance and impaired insulin secretion [58]. In addition to genetic factors, epigenetic changes play a critical role in the T2D development [59]. Chromatin modification, such as methylation, acetylation, and post-translational modifications to histone proteins is crucial for epigenetic regulation of genes involved in the development of T2D and obesity [60], which explains the activation of chromatin modifying enzymes genes.

Artificial intelligence is increasingly used in various spheres of life and may find applications in healthcare to assist in the diagnosis and treatment of different diseases, including T2D. The possibility of a machine learning model’s implementation in clinical practice for the T2D onset prediction and high-risk patients’ identification is now widely discussed. The operating principle of machine learning algorithms is based on the analysis of large amounts of data and the identification of patterns that can be used to make predictions. In the case of T2D, factors such as gender, age, BMI, laboratory values, as well as genetic and epigenetic indicators are typically analyzed. The predictive power of models for assessing polygenic risk score in multifactorial traits and disorders is currently quite low. The relative performance of the models varies from 0.571 to 0.901 [61]. The constructed machine learning model for prediction of T2D based on the expression levels of four miRNAs (miR-5588-5p, miR-125b-2-3p, miR-1284, and miR-496) showed relatively high performance (AUC = 0.94). However, the limited size of the training sample in this study makes it necessary to analyze the expression of the detected miRNAs on an expanded sample with the correction of the model, which can affect its performance. Adjusting the model for additional factors such as gender, age, BMI, genetic predisposition can also potentially increase the predictive power of the constructed model, since the influence of these factors on T2D susceptibility is well known [62,63,64,65]. 

Based on the results of the present study, four differentially expressed microRNAs were detected that were not previously associated with T2D by other researchers. There are a number of causes for impacting the expression profile and discrepancies in experiment results. For example, changes in miRNA expression can be altered by external factors such as clinical course of a disease, its complications (diabetic nephropathy, retinopathy) [66], and presence of concomitant diseases. As well, researchers demonstrated differences in miRNA expression in different populations [67]. In addition, the expression of a particular miRNA can change to the same extent in other pathological conditions. All these factors make extremely important the rigorous selection of study participants. Moreover, the analyzed biomaterial, sample handling, miRNA isolation method and conditions, preferential miRNA extraction or degradation during storage, and other technical variations (“batch effects”) [68] can also potentially affect the differential expression analysis results. One of the possible solutions to these problems is to search for a set of miRNAs with simultaneous expression changes, which is only possible when using methods such as NGS or microarrays.

The choice of expression analysis method is another significant issue. While hybridization arrays make it possible to analyze a larger number of miRNAs compared to RT-qPCR, and NGS even allows identifying new ones, it is necessary to verify the results of these studies using RT-qPCR. Unfortunately, the verification step is often missing, which complicates the selection of miRNAs for validation of obtained results on an extended sample. In our study, differential expression for miR-5588-5p, miR-125b-2-3p, miR-1284, and miR-496 was verified using RT-qPCR; however, significant differences were demonstrated only for miR-496, which is likely due to the small sample size.

The limitations of this study, in addition to the limited sample size, include the lack of validation of the results on an expanded sample, as well as differences in age, WHR, and BMI between T2D patients and healthy controls. Lack of validation can be overcome through the additional research conduction; however, age, hypoglycemic medications, WHR, and BMI cannot always be considered while conducting a study on T2D since classic manifestations of T2D develop in older people, and increased BMI and abdominal obesity are two of the main risk factors for T2D and a consequence of insulin resistance at the same time. In addition, increased WHR and BMI are a limitation that is difficult to overcome since it is not always amenable to therapeutic correction for various reasons, ranging from resistance to therapy to low compliance with therapy and recommended diet. The selection of subjects for the control group was based on the minimum risk of developing T2D and, as a result, the absence of increased WHR, BMI, and older age. 

Further studies are required to investigate the functions of analyzed miRNAs and their involvement in T2D pathogenesis, which have not yet been evaluated in T2D patients. RT-qPCR of detected miRNAs in an expanded sample, transcriptomic analysis in T2D patients for identification of target genes followed by RT-qPCR verification may help shed some light on the pathogenesis of T2D. Another important direction in miRNAs studies is the consideration of detected miRNAs as a potential biomarker for presymptomatic diagnosis of T2D. To implement this direction, it is necessary to analyze the expression of miRNAs in a large sample, considering associated factors, as well as long-term dynamic monitoring of miRNA expression in patients and healthy controls. Another promising direction for future research is single-cell transcriptomic studies performed on islet cells and peripheral blood mononuclear cells, as they will give us a better understanding of proliferation and differentiation of islet cells, as well as pathogenesis of immune inflammation at T2D. 

## 4. Materials and Methods

### 4.1. Study Cohorts and Participants

Peripheral blood samples were collected from 22 healthy controls and 44 individuals with T2D. T2D was diagnosed based on the World Health Organization criteria. The inclusion criteria for the control group were no history of diabetes and age over 30 years. Patients with new-onset diabetes mellitus (less than 1 year), acute and/or decompensated liver and kidney disease, autoimmune disorders, malignancies, and under 30 years of age were excluded from the study. The levels of HbA1c, fasting blood glucose (FBG), high-density lipoprotein (HDL), low-density lipoprotein (LDL), total cholesterol, and creatinine were measured in a fasting blood sample. Body height and body weight were measured, and the body mass index (BMI) was calculated in all the participants. Clinical and biochemical parameters of all the participants enrolled in the study are summarized in Table 2.

The study was approved by the ethics committee of the D.O. Ott’s Institute of Obstetrics, Gynecology, and Reproductology (protocol #130 dated 16 July 2020) and conducted in accordance with the guidelines of the Declaration of Helsinki. Patients with T2D and healthy controls were recruited at the D.O. Ott Institute of Obstetrics, Gynecology, and Reproductology (Saint-Petersburg) and St. Martyr George City Hospital (Saint-Petersburg). Written informed consent was obtained from all the subjects for being included in the study. This study was performed using large-scale research facilities #3076082 “Human Reproductive Health”.

### 4.2. Plasma Sample Collection

Whole blood samples were collected from the study subjects in Improvacuter 9 mL K2EDTA tubes (Guangzhou Improve Medical Instruments Co., Ltd., Guangzhou, China) after an overnight fasting period of 12 h. The plasma was separated by centrifugation of the whole blood sample at 1500 g for 10 min at room temperature. The plasma samples were then immediately aliquoted in RNAse-free cryotubes (Fluidx Ltd., Cheshire, UK) and stored at −80 °C until use to prevent freeze–thaw cycles.

### 4.3. Small RNA Isolation

Archived plasma specimens were retrieved from cryostorage and thawed at +4 °C. The total RNA, including miRNA, was isolated from 200 μL of thawed plasma using miRNeasy Serum/Plasma Advanced Kit (Qiagen GmbH, Germany) following the manufacturer’s instructions, eluted with 20 μL RNase-free water, and quantified with the use of Qubit™ microRNA Assay Kit and Qubit 2.0 Fluorometer (both Invitrogen™, Carlsbad, CA, USA). The miRNA concentration ranged from 0.7 to 3.8 ng/μL. MiRNA samples were stored at −20 °C until library preparation.

### 4.4. Small RNA Libraries Preparation and Sequencing

Libraries were constructed from 5 μL of extracted small RNA using QIAseq miRNA Library Kit (Qiagen GmbH, Hilden, Germany) according to the manufacturer’s protocol. Briefly, reverse transcription was performed after adapter ligation to the 3′ and 5′ ends of mature miRNAs. Following cDNA cleanup, library amplification, unique index assignment, and final cleanup were completed. The concentration of libraries was measured using Qubit dsDNA HS Assay Kit (Invitrogen™, CA, USA). The library’s size distribution was analyzed with Agilent 2200 TapeStation and Agilent High Sensitivity D1000 ScreenTape (both Agilent Technologies, Inc., Waltham, MA, USA). Barcoded libraries were pooled at equimolar ratios and paired-end sequenced with Illumina HiSeq 2500 System using HiSeq Rapid SBS Kit (all Illumina, Inc., Waltham, MA, USA).

### 4.5. Differential Expression Analysis

Small RNA-seq data alignment and quantification was conducted using GeneGlobe Data Analysis Center (https://geneglobe.qiagen.com/us/analyze, accessed on 2 November 2020). The analysis of miRNA differential expression (DE) was performed using the UMI counts. The DESeq2 [69] package was used to conduct the DE analysis. Adjusted *p*-value (<0.05) was used to select differentially expressed transcripts, and logarithm of fold expression change (|log2FC| > 1) was used to further filter the set of candidates.

Selection of candidate miRNA for further validation was carried out based on adjusted *p*-value, expression level, and the proportion of samples with zero expression. Combined relevance of miRNAs selected for validation was tested by fitting a support vector machine (SVM) model to predict the patient’s phenotype based on the expression of the miRNA candidates. The area under the receiver–operator curve (ROC/AUC) was used to evaluate the model performance in three-fold cross-validation. The model was constructed and evaluated using the caret package.

### 4.6. Functional Analysis of miRNA Target Genes

To select the target genes of the selected subset of miRNA, we used miRTarBase (https://mirtarbase.cuhk.edu.cn/, accessed on 10 December 2022). Unique target genes were selected for further functional analysis. Gene set enrichment analysis on the set of miRNA target genes was performed using the clusterProfiler package [11]. Gene Ontology (GO) biological process terms and Molecular Signatures Database (MSigDB) [70] canonical pathways were used as target gene sets for analysis. FDR-adjusted *p*-value < 0.1 was used to identify significantly enriched gene sets.

### 4.7. Quantitative Real-Time Polymerase Chain Reaction (RT-qPCR)

The results obtained by RNA-seq were verified through reverse transcription-quantitative PCR. 20 RNA samples (10 T2D and 10 controls) from those previously analyzed by NGS were randomly selected for the verification. Prior to RT-qPCR, cDNA was synthesized using TaqMan Advanced miRNA cDNA Synthesis Kit (Thermo Fisher Scientific, Waltham, MA, USA). The concentration of the resulting cDNA was measured using a NanoDrop 2000 spectrophotometer (Thermo Fisher Scientific, Waltham, MA, USA) and ranged from 120 to 130 ng/μL. Candidate miRNAs were selected for RT-qPCR verification based on fold changes and *p* values. PCR was performed using commercially available TaqMan Advanced miRNA Assays for measurement of miR-361-5p, miR-496, miR-5588-5p, miR-125b-2-3p, and miR-1284 expression levels (purchased from Thermo Fisher Scientific, Waltham, MA, USA). The reaction mixtures (20 μL) consisted of TaqMan Advanced miRNA Assays (1 μL), TaqMan Advanced miRNA Master Mix (10 μL), nuclease-free water (4 μL), and cDNA product (5 μL). The amplification was performed in MicroAmp optical reaction plates using ABI 7500 Fast Real time PCR system (all Applied Biosystems, Waltham, MA, USA) with two technical replicates for each PCR reaction. The RT-qPCR temperature cycling conditions were as follows: initial denaturation 95 °C for 20 s, followed by 40 cycles (95 °C for 3 s and 60 °C for 30 s). MiR-361-5p was selected as an endogenous control on the recommendation of the assay’s manufacturer as a gene with relatively stable expression across samples and tissues. Results were analyzed using comparative delta Ct (2^−ΔΔCt^) method [71].

### 4.8. Statistical Analysis

Statistics were calculated using Statistica 12.0 software (Tibco, Palo Alto, CA, USA). The Mann–Whitney U test and Fisher’s exact test were used to compare differences between groups. Differences were considered significant at *p*-value < 0.05. The sample size was calculated considering the prevalence of T2D in the study population, equal to 3.3% [72], the 80% sample size power at 95% confidence interval, using formula [73].

### 4.9. Machine Learning Model Construction

The combined relevance of miRNAs selected for validation was tested by fitting a support vector machine (SVM) model to predict the patient’s phenotype based on the expression of candidate miRNAs. The area under receiver–operator curve (ROC/AUC) was used for evaluation of the model performance in three-fold cross-validation. The model was constructed and evaluated using the caret package.

## 5. Conclusions

MicroRNA profiling studies look intriguing for opening new perspectives in fundamental research and for developing new instruments for T2DM diagnosis. According to modern knowledge and molecular characteristics of microRNAs, they can be considered excellent early or presymptomatic biomarkers due to their role in the initial stages of pathogenesis of many chronic diseases. This is especially important for T2DM, a serious chronic disease that can remain asymptomatic for a long time. Our preliminary results showed differential expression in plasma of miR-5588-5p, miR-125b-2-3p, miR-1284, and miR-496 in T2DM and miR-144-3p and miR-99a-5p in obesity. Functional analysis demonstrated the importance of cellular processes such as cell cycle regulation, apoptosis, and chromatin modifications in T2D development. Further validation of miRNAs’ differential expression on an extended sample using alternative techniques and analysis of target genes expression, protein synthesis, and organ functions may shed light on some aspects of the intricate structure of T2D pathogenesis. 

## Figures and Tables

**Figure 1 ijms-24-17406-f001:**
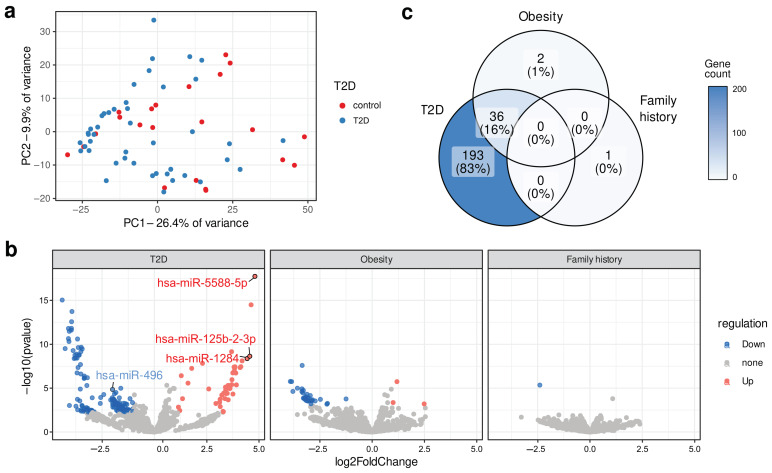
Differential miRNA expression analysis in Russian T2D patients. (**a**) Scatterplot showing the results of principal component analysis of rlog-transformed miRNA expression profiles for 22 controls and 44 T2D patients. (**b**) Volcano plot showing differentially expressed miRNAs for patients with T2D, obesity, and family history of T2D. (**c**) Venn diagram showing overlapping differentially expressed miRNAs for T2D, obesity, and family history of T2D.

**Figure 2 ijms-24-17406-f002:**
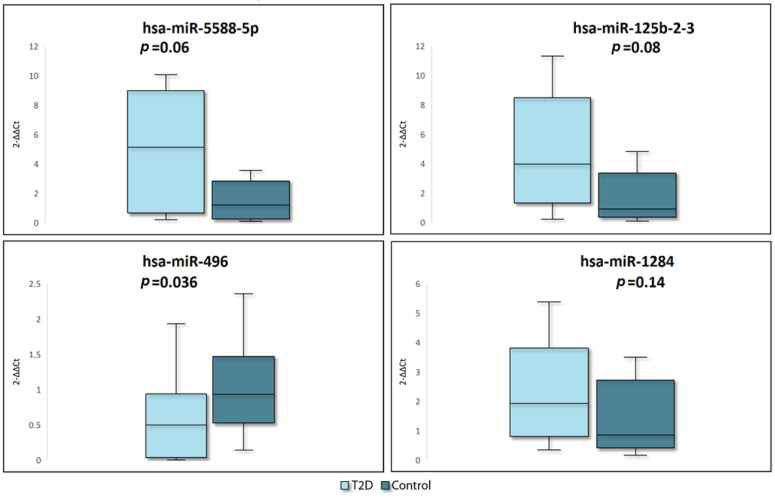
Verification of NGS results by RT-qPCR. The box represents the interquartile range (from the 25 to 75 percentile), the line inside the box represents the median, the whiskers represent the maximum and minimum values.

**Figure 3 ijms-24-17406-f003:**
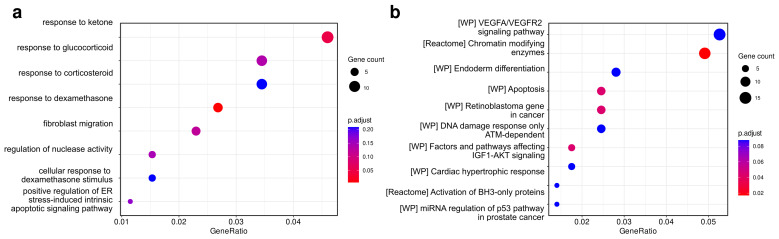
Dot plots showing the results of gene set enrichment analysis of target genes for the 4 identified miRNAs (miR-5588-5p, miR-125b-2-3p, miR-1284, and miR-496). Gene Ontology biological process (GO-BP) (**a**) and Molecular Signatures Database (MSigDB) canonical pathways (**b**) are used as target gene sets in the analysis.

**Table 1 ijms-24-17406-t001:** Description of detected miRNAs associated with T2D.

MiRNA	log2FC	Adjusted *p*-Value	Associated Conditions	Tissues/Biological Fluids	Pathways	References
miR-5588-5p	4.81	3.7 × 10^−15^	Alzheimer’s disease	Whole blood, serum, cerebrospinal fluid, plasma	Further studies are required	[28]
miR-125b-2-3p	4.43	3.1 × 10^−7^	Hepatocellular carcinoma, Ischemic stroke, colorectal cancer	Islet cells, visceral adipose tissue, PBMC, plasma	AMPK pathway	[29,35,36,37,38,39,40,41,42,43,44]
miR-1284	4.20	5.4 × 10^−7^	Breast, gastric, lung cancer cancer, osteosarcoma	Tumor tissues, plasma	PI3K/Akt, p27 pathways	[45,46,47,48,49,50,51]
miR-496	−3.90	4.4 × 10^−4^	Osteosarcoma, T2D	PBMC, plasma	PI3K/AKT, mTOR pathway	[30,31,32,33,34]

**Table 2 ijms-24-17406-t002:** Clinical and biochemical characteristics of T2D patients and healthy controls.

Characteristics	T2D Patients (n = 44)	Healthy Controls (n = 22)
Male (n)	14	11
Female (n)	30	11
Age (years)	73.15 ± 7.72 *	42.90 ± 16.52
FBG (mmol/L)	8.34 ± 2.64 *	4.62 ± 0.39
Family history of diabetes (n)	17 (38.6%)	8 (36.3%)
HbA1c (%)	8.36 ± 1.42	NA
HDL (mmol/L)	1.29 ± 0.33 *	1.60 ± 0.54
LDL (mmol/L)	2.62 ± 0.88	3.35 ± 1.69
Total cholesterol (mmol/L)	4.59 ± 1.14	5.57 ± 1.79
Creatinine (mmol/L)	0.115 ± 0.03 *	0.083 ± 0.01
BMI (kg/m^2^)	30.25 ± 4.98 *	23.98 ± 2.23
WHR	0.95 ± 0.08 *	0.75 ± 0.09

* *p* < 0.05 compared to control group. NA: not applicable. The level of glycated hemoglobin was not assessed in the control group.

## Data Availability

The datasets used and/or analyzed in the present study are available from the corresponding author upon a reasonable request. All code and intermediated data files pertinent to the analysis described in this work are available in the repository at https://github.com/mrbarbitoff/t2d_mirna/ (accessed on 5 December 2023).

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
