# Peer review of "Plasma microRNA Profiling in Type 2 Diabetes Mellitus: A Pilot Study"

_ijms, 2023, doi:10.3390/ijms242417406_

Round 1
Reviewer 1 Report
Comments and Suggestions for Authors
The aim of the present study was to analyze the spectrum of differentially pressed miRNAs in the blood plasma of T2D patients and assess their functional role in evaluating the prospects in the use of these miRNAs as potential biomarkers for risk stratification and T2D pre-symptomatic testing.
Comments
1. I am afraid that there is no conclusion in this study. What were clinical implication or biomarkers of specific miRNAs?
2. Characteristics of patient with T2D or healthy control were not presented at all. The heterogeneity of T2D introduces confusion and possible confounding.
3. Sample size calculation should be presented in the section of statistical analysis.
Reviewer 2 Report
Comments and Suggestions for Authors
The article by Ziravard N. Tonyan et al. presents an intriguing piece of work, with a well-structured experimental design and noteworthy results. The exploration and identification of new biomarkers for diabetes mellitus is a hot topic researched area.
However, two aspects merit further scrutiny:
1. The size of the analyzed population is relatively small for this type of investigation on miRNA. Consequently, the conclusions should be approached with caution, considering them more as a preliminary working hypothesis than definitive findings.
2. Including information about the sample power is crucial for establishing the statistical significance of the results.
3. In methods section, additional details regarding the utilization of artificial intelligence in data analysis would enhance the comprehensibility of the study.
Comments on the Quality of English LanguageDear Editor,
The English language appears to be satisfactory; there are only minor editing mistakes.
